# TinyAct: A framework for real-time action recognition in the cloud through distillation learning

Yupaporn Wanna, Kannika Wiratchawa, Thanapong Intharah◯*

Visual Intelligence Laboratory, Department of Statistics, Faculty of Science, Khon Kaen University, Khon Kaen, Thailand

* thanin@kku.ac.th

## Abstract

Human action recognition has become increasingly important for applications in security surveillance, healthcare monitoring, and smart environments. However, existing deep learning models typically require substantial computational resources, making deployment on resource-constrained edge devices challenging. To address this limitation, we propose TinyAct, a lightweight framework for real-time human action recognition that combines edge computing with cloud-based processing through knowledge distillation. TinyAct employs a 3D video autoencoder to extract compact spatiotemporal features from video sequences, coupled with classical machine learning classifiers for action prediction. The framework utilizes an AIoT (Artificial Intelligence of Things) architecture where feature extraction occurs on edge devices while classification is performed in the cloud, enabling real-time processing with reduced bandwidth requirements. To enhance performance, we implement knowledge distillation using the ILA-ViT-B/16 transformer as a teacher model to transfer temporal knowledge to our compact student architecture. Our experiments on the Kinetics-400 dataset demonstrate that TinyAct achieves competitive performance while maintaining computational efficiency. Using 16-frame video clips with 1024-dimensional latent features, Random Forest achieved the highest baseline accuracy of 57.00%, followed by SVM (55.00%) and XGBoost (54.00%). The autoencoder-based feature extraction significantly reduces computational overhead compared to end-to-end deep learning approaches while preserving essential spatiotemporal information for accurate action recognition. The knowledge distillation experiments reveal that training configuration critically affects performance, with non-pretrained student models achieving better results (15.11% with SVM) than pretrained ones under teacher supervision. This suggests that joint optimization of the encoder and classifier is essential for effective knowledge transfer in resource-constrained settings. TinyAct's modular architecture enables flexible deployment across diverse hardware configurations, supporting both lightweight edge inference and cloud-based training pipelines. The framework

**Data availability statement:** The Kinetics-400 dataset used in this work is publicly available at https://github.com/cvdfoundation/kinetics-dataset.

**Funding:** This study was financially supported by the Fundamental Fund of Khon Kaen University (fiscal year 2024, project number 200553) and the National Science, Research and Innovation Fund (NSRF), Thailand. The funders had no role in study design, data collection and analysis, decision to publish, or preparation of the manuscript.

**Competing interests:** NO authors have competing interests.

demonstrates that effective human action recognition can be achieved without computationally intensive deep networks, making it suitable for smart surveillance systems, IoT applications, and embedded devices where computational resources are limited.

## Introduction

Human action recognition from video [1] has emerged as a fundamental function in security and surveillance, healthcare monitoring, smart living environments, and human-computer interaction [2,3]. By enabling machines to automatically detect and interpret human movements, such systems offer essential capabilities for early anomaly detection, patient behavior analysis, and interactive AI systems.

Recent advances in deep learning, especially convolutional neural networks (CNNs), recurrent models (e.g., LSTM), and transformer-based architectures, have enabled substantial improvements in action recognition performance by modeling complex spatiotemporal dynamics [1,4]. However, these high-performing models often demand significant computational resources, making them impractical for deployment on resource-constrained environments such as mobile devices, embedded systems, and smart cameras.

Despite the increasing need for on-device intelligence, deploying real-time action recognition models at the network's edge remains a significant challenge. Hardware constraints such as limited memory, CPU/GPU capabilities, battery life, and latency requirements inhibit the use of large-scale systems in real-world edge scenarios [5,6]. Prior solutions have proposed lightweight CNNs and federated learning strategies to address part of this challenge [7–9], yet these methods often neglect the temporal dependencies critical to robust action understanding.

To bridge this gap, we propose *TinyAct*, a compact yet effective framework tailored for real-time human action recognition on edge devices. TinyAct uses a lightweight autoencoder to extract spatio-temporal features that are then fed into an efficient classifier. Crucially, we incorporate a *Knowledge Distillation* paradigm, using the high-performing transformer-based model ILA [10] as a teacher to transfer temporal knowledge to TinyAct's compact student model. This enables TinyAct to balance performance and resource efficiency without sacrificing accuracy.

We evaluate our approach on the Kinetics-400 dataset. Besides accuracy gains, TinyAct also supports privacy-aware deployment by limiting dependency on cloud resources.

In summary, the contributions of this paper are:

1. We introduce *TinyAct*, a lightweight action recognition framework optimized for real-time inference on edge devices, leveraging knowledge distillation from a transformer-based teacher.

2. We design TinyAct's architecture to meet stringent hardware constraints while maintaining strong spatiotemporal modeling capabilities.

3. We conduct comprehensive experiments on Kinetics-400, showing TinyAct achieves performance comparable to larger models with far fewer resources—making it ideal for deployment in smart surveillance, IoT, and embedded systems.

The remainder of this paper is organized as follows. The Related Work section reviews related work on lightweight architectures, real-time action recognition, and knowledge distillation. The Methodology section presents the TinyAct methodology, including the AIoT architecture, autoencoder design, and knowledge distillation framework. The Results section describes our experimental setup and presents results on the Kinetics-400 dataset. The Discussion section discusses system scalability, privacy considerations, and the knowledge distillation performance gap. Finally, the Conclusion and Future Work section concludes the paper and outlines future research directions.

## Related work

The related literature is categorized into three main groups: (1) light-weight architecture, (2) real-time action recognition models, and (3) knowledge distillation learning.

### (1) Lightweight Neural Network Architectures for Edge Devices

A critical challenge in deploying deep learning models on edge devices is achieving a balance between computational efficiency and predictive performance. Numerous studies have proposed lightweight neural network architectures specifically designed for resource-constrained environments. These models aim to reduce computational load and memory usage while maintaining acceptable accuracy.

For instance, S. Yang *et al.* [7] introduced EdgeCNN, a convolutional neural network optimized for image classification on edge devices. By using smaller input sizes and efficient architectural designs—including quantization and transfer learning—EdgeCNN significantly reduces resource consumption. However, this model is tailored for static image classification and does not address the temporal complexity inherent in human action recognition tasks.

Y. Wu *et al.* [8] developed a low-light enhancement method to improve object detection performance using adaptive edge processing. While their work demonstrates the potential of edge computing for real-time object detection, it focuses on visual quality enhancement rather than action recognition.

In the domain of model optimization, P. Zhang *et al.*[11] employed federated learning and blockchain technology to enhance model performance in Industrial IoT settings, distributing training across edge devices. Similarly, U. De Alwis *et al.*[12] proposed TempDiff, a sparsity-inducing method for CNNs using temporal feature differences. Although effective in reducing resource usage, these techniques primarily address image-based or static tasks rather than video-based action recognition.

Some studies have also explored application-specific small networks. B. Hou *et al.*[13] introduced a deep reinforcement learning-based edge-cloud framework for real-time salient object detection. H. H. Nguyen *et al.*[14] adapted YOLO for fast human detection on edge devices, and R. Aarthy *et al.*[9] designed an edge-cloud collaboration framework for anomaly detection in video streams. Additionally, F. U. M. Ullah *et al.*[15] proposed an AI-powered edge vision system for real-time violence detection in industrial surveillance. Although these models demonstrate effectiveness in their specific domains, they often fall short in addressing the temporal dynamics required for human action recognition. TinyAct addresses this gap by leveraging autoencoders to extract features from video sequences and combining edge and cloud computing to enable real-time, low-latency action recognition.

Unlike prior approaches that focus on either static tasks or object detection, TinyAct is designed to handle complex temporal data while remaining computationally efficient for deployment on edge devices. Usman et al. [16] proposed a layer-frozen dual attention network for deepfake detection, combining frozen EfficientNetV2 features with Convolutional Block Attention Modules (CBAM) for channel and spatial attention. Their approach achieves competitive accuracy on multiple benchmarks including FaceForensics++, Celeb-DF, and DFDC, while reducing training cost through layer freezing. While their method targets spatial artifact detection in static images, TinyAct addresses temporal modeling in video

sequences. Both prioritize edge efficiency but optimize differently: deepfake detection requires fine-grained spatial analysis, whereas action recognition needs motion dynamics across frames. We use 3D autoencoders for temporal compression rather than spatial attention, reflecting these different task requirements.

Recent work has also emphasized the importance of developing feature representations that balance computational efficiency with privacy considerations in edge deployments. Khan et al. [17] demonstrated that selective attention mechanisms can extract robust, task-relevant features while maintaining resilience to environmental uncertainties. This principle extends naturally to privacy preservation: by focusing on action-discriminative patterns through knowledge distillation rather than preserving all visual details, compact feature representations can inherently emphasize task-relevant information over potentially sensitive identity-revealing features, providing baseline privacy protection in cloud-edge architectures.

Attention mechanisms have proven effective for lightweight architectures. Recent work on multi-scale trapezoidal attention [18] demonstrates efficient temporal feature extraction through hierarchical attention structures that reduce computational complexity compared to full self-attention. The trapezoidal design progressively narrows attention scope across layers, focusing computational resources on salient regions while maintaining global context. Applied to video, such mechanisms could identify motion-salient frames and spatial regions where actions occur. However, TinyAct prioritizes inference speed through direct feature compression via 3D autoencoders, reserving attention mechanisms as a future enhancement to balance accuracy gains with edge deployment constraints.

In summary, while prior research in lightweight neural networks has significantly improved the feasibility of deep learning on edge devices, most methods are limited to static tasks. TinyAct offers a comprehensive solution tailored for real-time human action recognition by integrating lightweight design, temporal modeling, and collaborative edge-cloud processing.

## (2) Real-Time Action Recognition in Video Analysis

Human action recognition in video is a core task in computer vision, with wide-ranging applications in surveillance, healthcare, autonomous systems, and human-computer interaction. Over time, research in this area has led to the development of models that aim to achieve high accuracy while supporting real-time processing. These models fall into three major categories: transformer-based architectures, self-supervised learning methods, and lightweight designs optimized for resource-constrained environments.

Transformer-Based Architectures for Action Recognition: Transformers have emerged as powerful tools for capturing temporal dependencies in video data. Tu et al.[10], which is designed to improve temporal modeling by learning flexible alignment across video frames. ILA achieves 88.7% top-1 accuracy and 97.8% top-5 accuracy on the Kinetics-400 dataset without using extra training data, demonstrating its robustness and generalization capability. The model is based on the ViT-B/16 backbone with 12 layers, a hidden size of 512, 12 attention heads, and operates on input clips of 16 frames with 224×224 resolution. In our work, we adopt ILA as the teacher model in the knowledge distillation process to guide the training of the TinyAct student model, transferring rich temporal understanding from a large transformer to a compact architecture suitable for edge devices.

Park et al. [19] proposed a dual-path adaptation framework that enhances transformer adaptation from image to video domains by combining spatial and temporal feature learning. While effective, such transformer-based approaches often require significant computational resources, making them less suited to edge deployment. TinyAct, in contrast, avoids the high resource cost of transformer architectures by leveraging autoencoders and efficient classifiers for lightweight temporal modeling.

Self-Supervised Learning for Action Recognition: Self-supervised models reduce the need for large labeled datasets by learning video representations through pretext tasks. One such example is VideoMAE [20], which employs masked autoencoding to learn spatiotemporal features from unlabeled video data. While this approach improves generalization and data efficiency, it relies on heavy pre-training and large-scale computation. TinyAct shares the use of autoencoders

but bypasses the complexity of self-supervised pretraining in favor of lightweight training pipelines designed specifically for real-time edge applications.

Lightweight Architectures for Efficient Temporal Modeling: Efficiency-focused models like MoViNets [21], TSM [22], and R(2+1)D [23] have shown promising results in real-time video understanding. MoViNets apply depthwise separable and temporal convolutions for mobile video recognition. TSM cleverly shifts temporal features to reduce the need for 3D convolutions, achieving impressive performance with minimal parameters. R(2+1)D decomposes 3D convolutions into spatial and temporal parts, reducing complexity while maintaining accuracy.

While these models prioritize efficiency, their reliance on specialized convolutions or architectural complexity may still limit their deployment on low-power edge devices. TinyAct addresses this gap by employing autoencoders to extract feature vectors from video sequences, reducing the input dimensionality and simplifying temporal modeling. This results in a significantly lower computational footprint while maintaining competitive accuracy.

## (3) Knowledge Distillation for Model Compression

Knowledge Distillation (KD) is a widely adopted technique for model compression, aiming to transfer knowledge from a large, powerful teacher model to a smaller, more efficient student model. Hinton *et al.* [24] first introduced the idea of using soft targets derived from the teacher's output distribution to guide the training of the student. This soft supervision allows the student model to capture richer information about class similarities and generalization behavior, leading to improved performance even with significantly fewer parameters. Gou *et al.* [25] surveyed a wide range of KD approaches and categorized them into three primary types: response-based, feature-based, and relation-based methods. For instance, the Fit-Nets approach [26] aligns intermediate representations of teacher and student networks using a hint-based loss function, while AT (Attention Transfer) [27] focuses on transferring attention maps from the teacher to the student. Other works such as RKD [28] and CRD [29] introduce relation-based and contrastive learning mechanisms to distill structural knowledge or instance-level relations between samples.

These advancements have made KD an essential tool in building deployable deep learning systems. Recent advances in video understanding have highlighted the importance of attention mechanisms for capturing temporal complexity. Khan et al. [30] proposed a hybrid attention-based approach that combines channel-wise and spatial-wise attention to refine temporal features in video summarization, demonstrating that adaptive feature weighting can effectively highlight discriminative temporal patterns. Drawing from these principles, TinyAct's knowledge distillation approach leverages multi-level temporal supervision: the ILA teacher provides rich temporal guidance through its learnable alignment mechanism, while the student's autoencoder-MLP architecture efficiently compresses and classifies temporal patterns via 3D convolutions and residual layers. This strategy enables effective temporal knowledge transfer to compact student architectures for edge deployment, where explicit attention modules would be computationally prohibitive. In edge computing scenarios, where latency and memory usage are critical, KD enables compact models to retain the expressive power of large networks. Inspired by this principle, TinyAct employs KD to inherit rich temporal features from a transformer-based teacher model (ILA), allowing it to operate effectively in real-time on resource-constrained edge devices.

While state-of-the-art models in action recognition—such as transformer-based architectures (e.g., ILA [10]), self-supervised learning approaches (e.g., VideoMAE [20]), and lightweight designs (e.g., MoViNets [21], TSM [22])—have demonstrated impressive performance, most of them are constrained by either high computational requirements or complex pretraining pipelines. These factors limit their practicality for real-time deployment in edge computing environments. TinyAct addresses these limitations by focusing on model compression and computational efficiency while preserving strong performance in temporal modeling. Unlike transformer-based models such as ILA that require substantial GPU memory and inference time, TinyAct employs a compact architecture guided by knowledge distillation. By transferring rich temporal features from a large teacher model (ILA) to a small student network, TinyAct retains essential capabilities for action recognition with significantly reduced resource demands. In contrast to self-supervised approaches that require

extensive pretraining on large-scale video datasets, TinyAct avoids such overhead by using lightweight autoencoders for feature extraction and direct supervised learning. Compared to lightweight models like MoViNet or TSM, TinyAct further simplifies temporal modeling by leveraging autoencoders to compress temporal information early in the pipeline, leading to lower latency and improved adaptability on diverse edge devices. In summary, TinyAct bridges the gap between accuracy and efficiency by offering a real-time action recognition framework that is both lightweight and edge-ready—making it a practical solution for applications in smart surveillance, IoT systems, and embedded devices where computational constraints are critical.

## Methodology

### AIoT Architecture

The architecture of TinyAct is designed based on an AIoT (Artificial Intelligence of Things) paradigm, enabling real-time human action recognition while balancing computational efficiency and classification accuracy. As illustrated in Fig 1, the architecture is divided into two primary components: a) Edge device and b) Processing in the cloud.

### a) Edge device

The edge device is responsible for data collection, preprocessing, and feature extraction:

- **Data preprocessing**: The Load Data Module captures images from multiple CCTV cameras (e.g., CCTV_1, CCTV_2, ..., CCTV_5) in a round-robin manner every 10 seconds to ensure continuity while reducing bandwidth and processing overhead. The captured frames are grouped and sent to the encoder for further processing.

- **Encoder module**: A lightweight 3D autoencoder processes the video frames to generate latent representations $(z_1, z_2, \ldots, z_k)$ capturing spatio-temporall information. This module simultaneously reconstructs the input for validation and extracts compressed feature vectors, which are transmitted to the cloud for classification via a REST API (HTTP POST) in JSON format.

- **Web application**: A real-time monitoring interface displays live camera feeds and integrates prediction outputs from the edge and cloud. All captured data are stored locally in the edge device database, ensuring privacy-preserving data management while allowing users to access real-time insights within the local network.

### b) Processing on cloud

The cloud processing component manages action classification, storage, and visualization:

- **Action classification**: The cloud receives feature vectors via HTTP requests and performs action recognition using the TinyAct model, classifying human activities across multiple categories in real-time.

- **Cloud database**: Classification results, including camera ID, timestamp, and predicted actions, are stored in a cloud database while also being sent back to the edge device database for local display.

- **Data extraction**: Structured data from the cloud database are extracted for further analysis, enabling pattern discovery and performance evaluation across various surveillance or smart environment deployments.

- **AI visualization**: A dashboard (e.g., Looker) presents results in the form of charts, graphs, and tables, providing an intuitive overview for operators to monitor activities, detect anomalies, and perform advanced analytics.

In summary, the TinyAct AIoT architecture(Fig 1) performs feature extraction on edge devices while leveraging cloud resources for classification and visualization. The modular design supports 5–7 concurrent camera streams through GPU-parallelized batch inference [$K$, 16, 3, 224 × 224], achieving ~3.2× throughput improvement over sequential

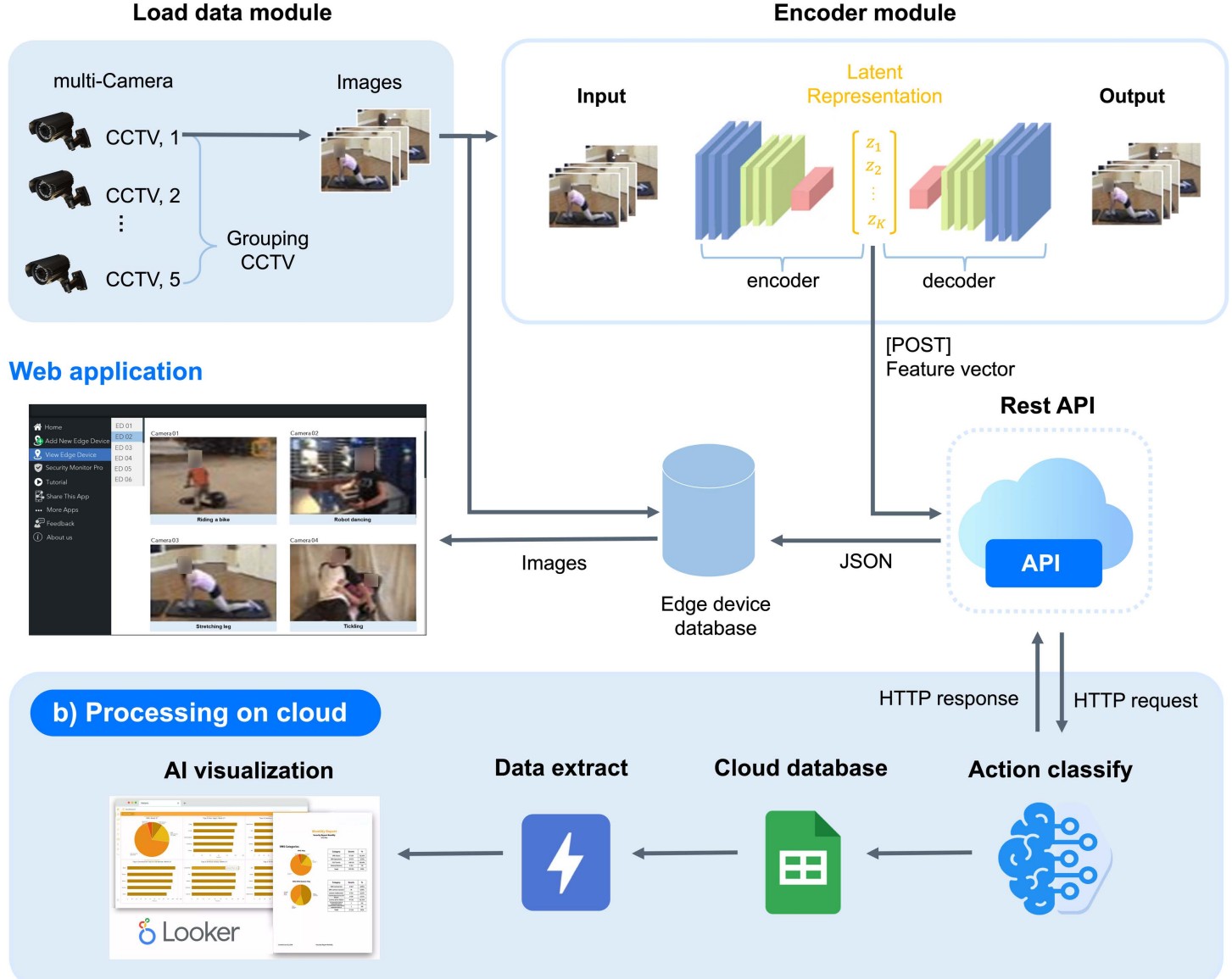

**Fig 1. TinyAct AIoT architecture.** Edge devices extract 1024-dimensional latent features via a lightweight 3D autoencoder and transmit them to the cloud for action classification and visualization. Note: Video frames are illustrative samples from the Kinetics-400 dataset [31]; facial regions have been obscured.

processing. The round-robin collection strategy (every 10 seconds per camera) distributes processing load temporally, while compressed feature transmission (1024-dimensional vectors=4KB per clip, 78–99× smaller than raw video) enables real-time operation over moderate-bandwidth connections (1–10 Mbps per stream). For deployments exceeding

single-device capacity (>7 streams), the architecture scales horizontally by deploying multiple edge devices with a local aggregation server coordinating cloud communication.

## Dataset

The Kinetics-400 dataset [31] is a large-scale video benchmark commonly used for action recognition research, consisting of YouTube video clips covering 400 human action classes with at least 400 clips per class. Each clip lasts approximately 10 seconds. We select Kinetics-400 for our experiments as it is a well-established benchmark with the fewest classes compared to newer versions (Kinetics-600, Kinetics-700). We use the standard train/test/validation split: 246,534 training clips, 19,906 validation clips, and 39,805 test clips. Before usage, we preprocess each set by sampling non-repeating data for each action equally. After this data management, the minimum number of actions in the train set is 228 for "taking a shower" class, 46 for "zumba" class in the validation set, and 93 for "dying hair" class in the test set, resulting in 91,200, 37,200, and 18,400 clips for train, validation and test data, respectively. These sets are used to develop a feature encoder and action recognition model.

The video frames depicted in Fig 1 are illustrative samples from this dataset, used solely for technical illustration of the system architecture. No human subjects were enrolled, observed, or recorded as part of this research. As a precaution-ary measure to comply with open-access publication requirements, facial regions in Fig 1 have been obscured to prevent potential identification of individuals appearing in the dataset samples.

### Base autoencoder for standalone action feature learning

The primary objective of the base autoencoder in the TinyAct framework is to serve as a foundational component for extracting high-quality, compact representations of spatio-temporal information from video sequences. Specifically, the autoencoder is designed to transform raw input frames into a lower-dimensional latent vector that encapsulates the essential features of human actions while significantly reducing the amount of data required for downstream processing. At this stage of the system development, the base autoencoder is trained in an unsupervised manner and operates inde-pendently from any knowledge distillation framework. It does not leverage a teacher model or soft labels; instead, it learns to encode the input data by minimizing the reconstruction error between the input video frames and their reconstructed outputs. This process allows the model to discover meaningful patterns inherent in the data, which can later be utilized by lightweight classifiers for action recognition. In summary, the base autoencoder acts as a general-purpose feature extractor that enables scalable and efficient deployment of action recognition tasks in resource-constrained environments, laying the groundwork for further enhancements such as knowledge distillation in subsequent phases.

### Architecture of the autoencoder model

The model is composed of two primary components: the encoder and the decoder.

**Encoder:** The encoder takes as input a sequence of video frames with a resolution of 256 × 256 pixels and a temporal length of $N$ frames ($N = 16$). It consists of multiple Conv2D layers interleaved with MaxPooling operations to progressively reduce the spatial dimensions while extracting hierarchical features. The output of the encoder is a latent representation vector, and we experiment with several latent dimensions: 1024.

**Decoder:** The decoder reconstructs the original video input from the latent vector using a sequence of UpSampling2D layers followed by Conv2D layers with ReLU activation. This design ensures that the encoder learns to preserve essential spatial and temporal features necessary for accurate reconstruction.

The full structure of the autoencoder and its integration with traditional machine learning classifiers is illustrated in Fig 2. As shown, the latent representation is not only used for reconstruction but also serves as the input to classifiers such as Support Vector Machines (SVM), Random Forest (RF), and Extreme Gradient Boosting (XGBoost) for down-stream action recognition.

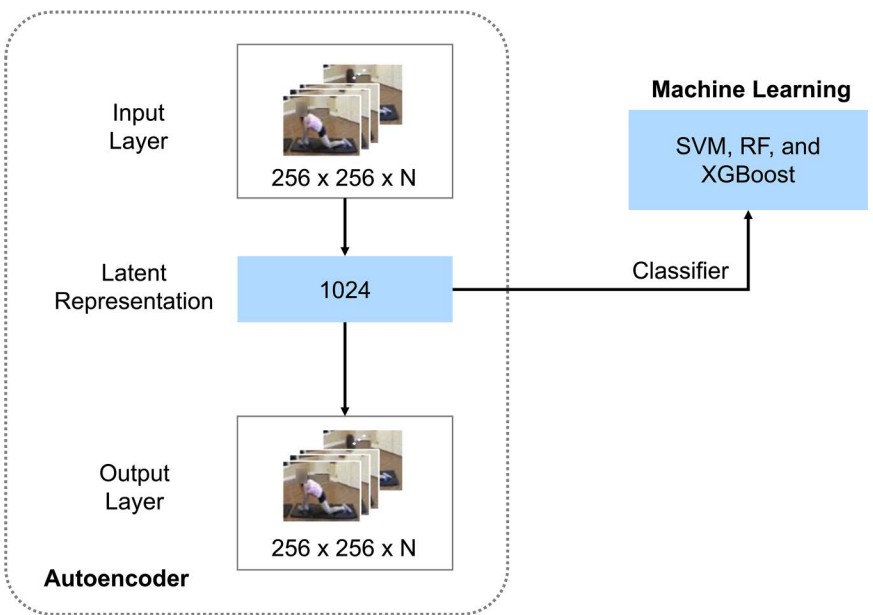

**Fig 2. The base autoencoder architecture showing the input and output layers with video frames of size 256×256×N, and the latent representation layer used as input for traditional machine learning classifiers including SVM, Random Forest, and XGBoost.**

## Training configuration and implementation details

To train the base autoencoder, we utilized the Kinetics-400 dataset, a widely used benchmark for action recognition tasks. This dataset consists of diverse human action clips collected from YouTube, spanning 400 action categories. For the training phase, we used a pre-processed subset comprising 91,200 clips, while the validation set contained 18,400 clips. The videos were sampled to ensure class balance and were resized to a fixed resolution of 256 × 256 pixels per frame.

During training, two different temporal window sizes were experimented with: 16 frames per video clip. These configurations were selected to study the effect of temporal information on the quality of the learned representations.

The autoencoder was trained in an unsupervised fashion using the Adam optimizer, which has been shown to perform well in deep learning tasks due to its adaptive learning rate and momentum-based updates. The objective was to minimize the Mean Squared Error (MSE) between the input video frames and their reconstructed outputs. This loss function measures the reconstruction quality and encourages the network to learn compact yet expressive latent representations.

The training process was conducted end-to-end, with the encoder learning to compress the spatio-temporal information into a latent vector, and the decoder learning to reconstruct the original input from this vector. No labels were used during this stage, as the autoencoder learns purely from reconstruction. The resulting latent representations were later evaluated in a supervised setting using traditional machine learning classifiers, as described in the next section.

## Layer-Freezing considerations for temporal feature learning

Following efficient architecture strategies for edge devices [16], we investigated layer-freezing during autoencoder training and knowledge distillation. Layer freezing—fixing pretrained network layers during training—reduces computational cost and has proven effective for image classification tasks.

Our knowledge distillation experiments compared two configurations: (1) pretrained autoencoder with frozen encoder layers, and (2) joint end-to-end optimization from scratch. Results show the frozen encoder achieved only 5.82% accuracy (SVM, Table 2) versus 15.11% for joint optimization. This performance gap reveals a key difference between spatial and

temporal learning. Unlike static image tasks where frozen features can be effectively reused, action recognition requires encoder-classifier co-adaptation to capture motion dynamics across frames. Simply freezing pretrained spatial features proves insufficient—the encoder must learn temporal compression that preserves motion-relevant information for downstream classification. This finding guided our training strategy: we use end-to-end optimization instead of layer freezing, allowing the model to adapt temporal feature extraction to action classification. This encoder-classifier co-adaptation is critical for preserving motion-relevant spatiotemporal information in the compressed 1024-dimensional representation, as fixed spatial features from unsupervised reconstruction cannot capture task-specific temporal dependencies required for discriminative action recognition. Future work could explore progressive unfreezing or intermediate layer distillation to better leverage pretrained temporal features.

### Utilizing latent representations in machine learning classifiers

After training the base autoencoder and obtaining latent vectors from the encoder, we evaluated the quality of these representations by using them as input features for a series of traditional machine learning classifiers. This step aimed to assess how well the unsupervised autoencoder captured discriminative features relevant to human action recognition.

Each video clip was passed through the encoder to generate a *latent vector* of fixed dimensionality—either 1024 depending on the autoencoder variant. These vectors served as input to the following classifiers:

- **Support Vector Machine (SVM):** A supervised learning model effective for high-dimensional data and widely used in classification tasks.

- **Random Forest (RF):** An ensemble method based on decision trees, known for its robustness to overfitting and strong performance on tabular data.

- **Extreme Gradient Boosting (XGBoost):** A gradient boosting framework that builds decision trees sequentially and optimizes performance through boosting techniques.

Each classifier was trained in a multi-class classification setting to predict one of the 400 action classes in the Kinetics-400 dataset. Hyperparameter tuning was performed using grid search and 10-fold cross-validation to ensure reliable evaluation. The best-performing models were selected based on classification accuracy on the validation set. The results demonstrate that the latent vectors extracted by the base autoencoder are sufficiently rich to enable high classification accuracy, particularly when using larger latent dimensions and longer input sequences (16 frames with 1024-dimensional vectors). This validates the effectiveness of the autoencoder as a standalone feature extractor, even before applying knowledge distillation.

## Knowledge distillation

### Teacher–student framework

To reduce the computational cost of large-scale video models while maintaining competitive performance, we adopt a knowledge distillation (KD) strategy in a teacher–student framework. In this setup, a powerful pretrained model that has demonstrated strong performance on standard video recognition benchmarks acts as the teacher, and a lighter model is trained to mimic its outputs. Specifically, our framework employs the ILA-ViT-B/16 model [10] as the teacher, leveraging its implicit temporal modeling via learnable alignment (ILA) to serve as a rich source of spatiotemporal supervision.

The teacher model, ILA-ViT-B/16, is based on CLIP's vision transformer backbone with 12 layers and a patch size of 16×16. It takes input videos consisting of 16 RGB frames, each with a spatial resolution of 224×224. The model has a hidden dimension of 768 and approximately 200 million parameters, enhanced with Implicit Spatio-Temporal attention (IST) blocks, which eliminate the need for explicit temporal attention by aligning semantically relevant regions across adjacent video frames. This enables efficient and effective temporal modeling. Pretrained on large-scale video-text data

(Kinetics-400 with CLIP initialization), the ILA teacher generates both logits for classification and embedding representations that capture mutual temporal information across frames.

To enable efficient video understanding under the knowledge distillation paradigm, we design a lightweight student model composed of two components: a 3D video autoencoder and a residual multi-layer perceptron (MLP) classifier. This architecture is tailored to process short-term video clips and distill rich supervision from a more complex teacher model, such as ILA-B/16.

## 3D Video autoencoder

The `VideoAutoEncoder3D` serves as the encoder–decoder backbone for extracting and reconstructing latent spatiotemporal representations from video inputs. It processes video tensors of shape [$B$, 16, 3, 224×224], where $B$ is the batch size, 16 denotes the number of temporal frames, and 224×224 represents the spatial resolution of each frame. The encoder comprises four sequential 3D convolutional blocks with progressively increasing channel depths (e.g., 64–512), employing strided convolutions to downsample both spatial and temporal dimensions. Each block is followed by batch normalization and ReLU activation to facilitate stable training and non-linear feature learning. The encoded output is aggregated via an adaptive average pooling operation, yielding a compact latent tensor of shape [$B$, 512, 1, 1, 1]. This is then flattened and linearly projected to a 1024-dimensional latent space, forming a semantic bottleneck that is shared across both reconstruction and classification objectives. The decoder symmetrically reverses the encoding process through a series of 3D transposed convolutions, gradually upsampling the latent representation to recover the original video structure. A final Sigmoid activation ensures that reconstructed pixel values lie within the normalized range [0, 1], thereby providing a reconstruction loss that acts as a regularization signal to enhance representation quality during training.

## Residual multi-Layer perceptron (MLP) classifier

Action classification is conducted via a dedicated residual multilayer perceptron (MLP) classifier, which processes the latent representation generated by the autoencoder. This classifier, referred to as `MLPResNetClassifier`, begins with a projection layer that maps the 1024-dimensional latent vector to a lower embedding dimension (typically 512), ensuring compatibility with the teacher model's feature space. The core of the classifier consists of a stack of $L$ repeated `MLPRes-Block` modules, where each block comprises a LayerNorm operation followed by two fully connected layers activated by GELU non-linearities, interleaved with dropout layers to mitigate overfitting. Each block is equipped with a residual skip connection to facilitate stable training and preserve gradient flow. Finally, the output head projects the normalized embedding into a vector of class logits corresponding to the target action classes (e.g., 400 for the Kinetics-400 dataset). In addition to classification, the architecture optionally exposes intermediate feature embeddings, which can be utilized for feature-level knowledge distillation objectives.

## Loss formulation in knowledge distillation

We adopt a composite loss function that integrates classification supervision, soft-target alignment, and reconstruction regularization to guide the training of the student model under a knowledge distillation paradigm, as illustrated in Fig 3.

Let the teacher network be represented by a feature extractor $f_t(\cdot; \Theta_t)$ and classifier head $g_t(\cdot; w_t)$, while the student network consists of an encoder $f_s(\cdot; \Theta_s)$ and classifier $g_s(\cdot; w_s)$. Given an input video sample $x_i \in \mathbb{R}^{T \times C \times H \times W}$ and its corresponding label $y_i \in \{1, \ldots, C\}$, the teacher produces logits $z_t = g_t(f_t(x_i))$, and the student yields logits $z_s = g_s(f_s(x_i))$. The softened class distributions are computed using the temperature-scaled softmax:

$$q_t^\tau = \text{softmax}\left(\frac{z_t}{\tau}\right), \quad q_s^\tau = \text{softmax}\left(\frac{z_s}{\tau}\right)$$

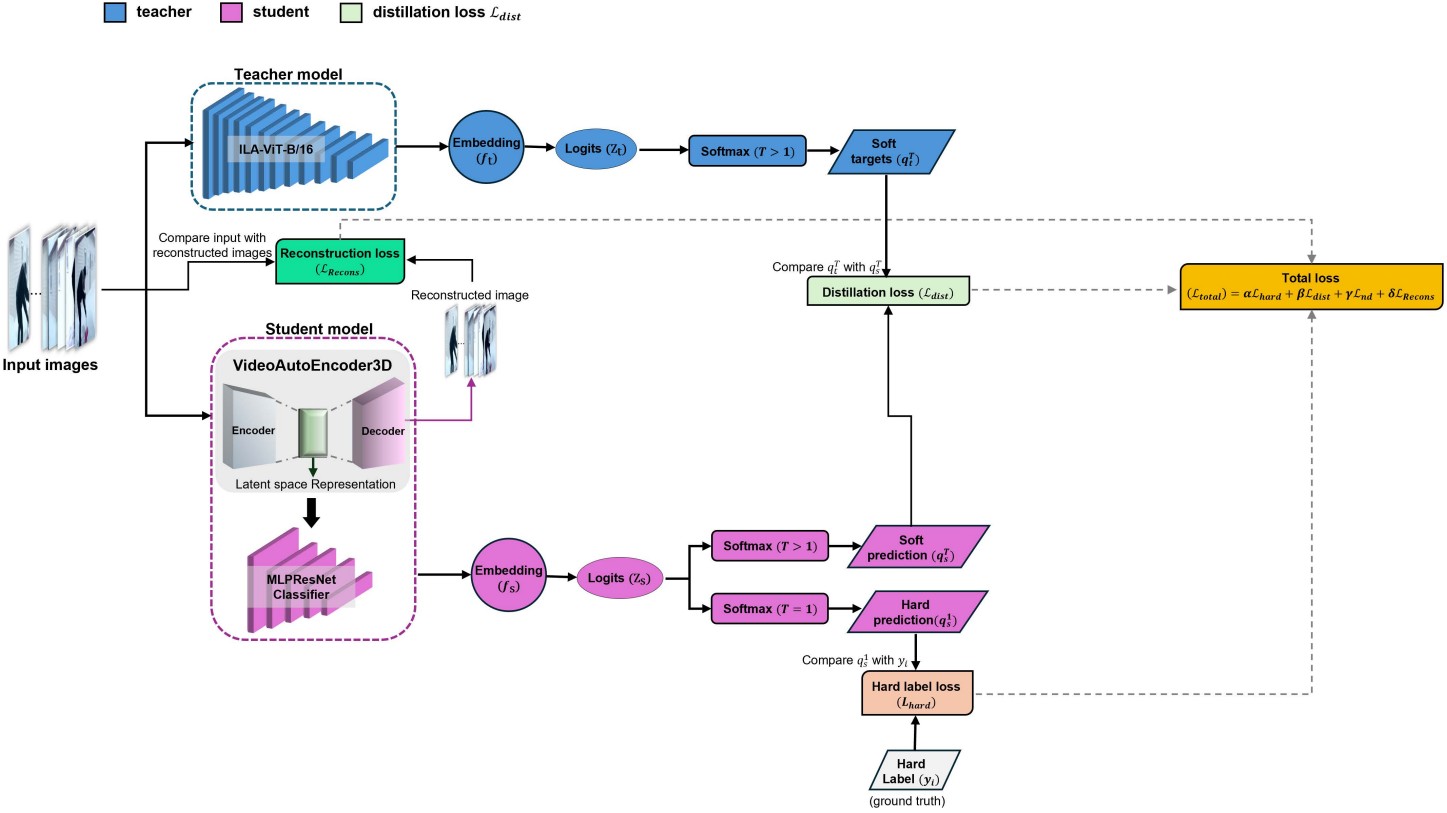

**Fig 3. An overview of the knowledge distillation framework employed in our training pipeline.** The student model—consisting of a `VideoAutoEncoder3D` coupled with `MLPResNetClassifier` is optimized to replicate the behavior of a pretrained teacher model (ILA-ViT-B/16) through the minimization of a composite loss function. This framework integrates three key components: (1) classification supervision via cross-entropy loss with ground-truth labels, (2) soft-target alignment by minimizing the Kullback–Leibler divergence between teacher and student logits, and (3) a reconstruction objective that serves as an auxiliary regularization signal to improve representational fidelity.

where $\tau > 0$ is the temperature parameter used to soften the output distributions. Our loss formulation includes the following components:

**Hard Label Loss** is a conventional cross-entropy loss or namely classification loss applied between the student's prediction and the ground truth:

$$\mathcal{L}_{\text{hard}} = \text{CE}(q_s^1, y_i)$$

**Distillation Loss** uses Kullback–Leibler (KL) divergence as the knowledge distillation (KD) loss between the softened output distributions of the teacher and the student, encouraging the student to mimic the teacher's output distribution. This formulation follows the seminal work by Hinton *et al.* [24], where knowledge from the teacher is transferred by minimizing the KL divergence between the temperature-scaled (softened) logits of the teacher and student:

$$\mathcal{L}_{\text{dist}} = T^2 \cdot \text{KL}(q_t^T \| q_s^T)$$

**Reconstruction loss (MSE or SSIM-based)** is applied between the input and the decoded output, since our student model incorporates an autoencoder. This loss serves as a regularization term that encourages the network to retain meaningful information during the encoding and decoding process:

$$\mathcal{L}_{\text{recon}} = \frac{1}{N} \sum_{i=1}^{N} \|x_i - \hat{x}_i\|^2$$

where $\hat{x}_i$ is the reconstructed input.

The total loss function used to train the student model is a weighted combination of the above objectives:

$$\mathcal{L}_{\text{total}} = \alpha \mathcal{L}_{\text{hard}} + \beta \mathcal{L}_{\text{dist}} + \delta \mathcal{L}_{\text{recons}}$$

### Handling temporal complexity in knowledge distillation

A critical challenge in applying knowledge distillation to action recognition lies in effectively transferring temporal knowledge from the teacher to the student model. Unlike image classification tasks where spatial features dominate, action recognition requires the model to capture complex temporal dependencies across video frames. To address this challenge, TinyAct employs several strategies inspired by recent advances in temporal feature extraction and hybrid attention mechanisms [30].

**Temporal Feature Alignment:** Our knowledge distillation framework explicitly addresses temporal complexity through multi-level feature alignment between teacher and student models. The ILA-ViT-B/16 teacher model captures temporal dependencies through its Implicit Spatio-Temporal attention (IST) blocks, which align semantically relevant regions across adjacent frames without requiring explicit temporal attention modules. The student model learns to replicate this temporal understanding by minimizing the distillation loss $\mathcal{L}_{\text{dist}}$ between softened logits, where the temperature scaling factor $\tau$ serves to expose the teacher's nuanced temporal reasoning across the 16-frame sequence. By softening the output distribution, the student gains access to the teacher's relative confidence across action classes at each temporal position, enabling more effective transfer of temporal discrimination capabilities.

**Progressive Temporal Learning:** Drawing on the principle that channel-wise attention emphasizes discriminative temporal patterns while spatial-wise attention preserves fine-grained motion details, we implement a progressive learning strategy where the student model first learns to extract frame-level spatial features, then gradually incorporates temporal dependencies through the distillation process. The 3D convolutional structure in our VideoAutoEncoder3D naturally preserves temporal ordering while reducing dimensionality—each Conv3D layer with kernel size (3, 3, 3) processes both spatial ($x$, $y$) and temporal ($t$) dimensions simultaneously, enabling the network to learn hierarchical temporal representations. This multi-scale temporal processing is analogous to the teacher's multi-layer temporal modeling, where early layers capture low-level motion patterns and deeper layers integrate these into high-level action semantics.

**Hybrid feature refinement for temporal patterns:** The temporal complexity in action recognition stems from both channel-wise and spatial-wise feature interactions across time. Channel attention can highlight discriminative temporal patterns by emphasizing feature maps that capture action-relevant motion, while spatial attention preserves fine-grained details within each frame. Our student architecture addresses this through a two-stage processing pipeline. The autoencoder encoder captures raw spatiotemporal patterns through 3D convolutions, yielding a 1024-dimensional latent vector $\mathbf{z} \in \mathbb{R}^{1024}$ that encodes compressed temporal information. Subsequently, the residual MLP classifier with LayerNorm and GELU activations learns to weight and refine these patterns based on the teacher's soft targets. Each MLPResBlock applies residual connections to preserve temporal information flow, maintaining long-range temporal dependencies. This design enables the compact student to focus on discriminative temporal features without the computational overhead of explicit attention modules.

**Reconstruction as temporal regularization:** The reconstruction loss $\mathcal{L}_{\text{recon}}$ serves a dual purpose in our framework: it acts as both a regularization term and a mechanism to preserve temporal coherence. By requiring the decoder to

reconstruct the original 16-frame sequence from the latent representation, we ensure that the encoder maintains temporal structure even under the influence of distillation supervision. Specifically, the MSE-based reconstruction objective

$$\mathcal{L}_{\text{recon}} = \frac{1}{N} \sum_{i=1}^{N} \|x_i - \hat{x}_i\|^2$$

enforces that the latent vector $z$ must retain sufficient spatiotemporal information to recover the input sequence $(x_1, x_2, \ldots, x_{16})$. This prevents the student from overfitting to the teacher's classification outputs while losing essential temporal patterns. The balance between classification accuracy (via $\mathcal{L}_{\text{hard}}$ and $\mathcal{L}_{\text{dist}}$) and temporal fidelity (via $\mathcal{L}_{\text{recon}}$) is controlled by the loss weights ($\alpha, \beta, \delta$), with our empirically determined configuration (0.5, 0.5, 0.5) achieving stable convergence and effective temporal knowledge transfer.

The coefficients $\alpha$, $\beta$, and $\delta$ are the weights for the hard-label supervision loss $\mathcal{L}_{\text{hard}}$, the distillation loss $\mathcal{L}_{\text{dist}}$, and the reconstruction loss $\mathcal{L}_{\text{recons}}$, respectively. We adopt the standard logit-based knowledge distillation (KD) method [24], in which $\mathcal{L}_{\text{dist}}$ is implemented as the KL divergence between the softened teacher and student outputs.

## Training configuration and implementation details

We train our knowledge distillation framework on the Kinetics-400 dataset, which comprises 400 categories of human actions. The teacher model is ILA-ViT-B/16 [10], a transformer-based architecture pretrained on Kinetics-400, which incorporates implicit temporal modeling via learnable alignment and processes 16-frame video clips of 224 × 224 spatial resolution. The student model is designed as a lightweight alternative, consisting of a 3D video autoencoder backbone that encodes short video segments into a 1024-dimensional latent representation, followed by a residual MLP classifier for action prediction.

Training is conducted over 100 epochs using stochastic gradient descent (SGD) with a momentum of 0.9, weight decay of $10^{-4}$, and an initial learning rate of $10^{-4}$, which is decayed to 1e-5 in the later stages. Input frames are normalized using ImageNet statistics [10], and each training sample has the shape [$B$, 16, 3, 224 × 224], where B denotes the batch size. The training pipeline is implemented using PyTorch in a distributed data-parallel setting across multiple GPUs.

The student model is supervised using a composite loss that integrates: (1) cross-entropy loss with ground-truth labels (weight $\alpha$ = 0.5), (2) Kullback–Leibler (KL) divergence between softened student and teacher logits (weight $\beta$ = 0.5), and (3) mean squared error (MSE) reconstruction loss between the input and decoded video frames (weight $\delta$ = 0.5). These weights were selected based on empirical tuning across a range of combinations to balance classification accuracy, distillation effectiveness, and reconstruction fidelity. The resulting configuration was found to offer stable convergence and competitive performance on downstream metrics. The distillation loss follows the seminal work of KD [24] where the teacher and student outputs are softened using a temperature scaling factor. We empirically assess temperature values $T \in \{1, 2, 4, 6, 8\}$, and find that $T = 4$ provides the most favorable trade-off between gradient informativeness and training stability. To mitigate early-stage instability, we apply a warm-up strategy during the first 20 epochs before activating the full KD loss weight. Throughout training, we track both classification accuracy and reconstruction quality to ensure the model retains semantic discriminability and spatial consistency.

## Temporal knowledge transfer strategy

The effectiveness of knowledge distillation in TinyAct critically depends on how temporal knowledge is transferred from teacher to student. We adopt a staged training approach to address temporal complexity in action recognition.

**Warm-up Phase:** During the first 20 epochs, the student learns basic spatiotemporal features through reconstruction loss ($\delta$ = 0.5) and ground-truth supervision ($\alpha$ = 0.5), while $\beta$ gradually increases from 0 to 0.5. This allows the 3D convolutional encoder to establish stable temporal representations— capturing frame-to-frame motion and short-term

dependencies—before mimicking the teacher's complex temporal modeling. Direct optimization with full distillation from the start causes unstable training, as the student attempts sophisticated temporal reasoning without basic encoding capabilities.

**Full Distillation Phase:** After warm-up, full distillation ($\beta = 0.5$) refines temporal understanding through teacher-student alignment. We selected $T = 4$ empirically from $T \in \{1, 2, 4, 6, 8\}$. Higher temperatures ($T \geq 6$) over-smooth temporal distinctions between similar actions, while lower temperatures ($T \leq 2$) provide insufficient smoothing, failing to transfer nuanced temporal patterns. At $T = 4$, softened distributions provide optimal gradient information for temporal learning.

**Joint Optimization:** Table 2 shows that non-pretrained students achieve 15.11% accuracy versus 5.82% for pretrained encoders (SVM classifier), revealing a key insight: fixed temporal features from unsupervised reconstruction cannot capture task-specific temporal dependencies. Pretrained encoders commit to representations optimized for reconstruction fidelity, emphasizing low-level motion patterns rather than discriminative action features. Joint optimization enables the student to learn representations serving dual objectives—reconstructible via $\mathcal{L}_{recon}$ yet discriminative via $\mathcal{L}_{hard}$ and $\mathcal{L}_{dist}$—balancing temporal coherence with classification utility for resource-constrained deployment.

## Evaluating the utility of student latent representations

After training the student model using knowledge distillation (KD), we obtained a 1024-dimensional latent vector from the encoder of the `VideoAutoEncoder3D`. This vector is learned with supervision from three loss functions: classification loss, distillation loss, and reconstruction loss. The vector is expected to contain meaningful spatiotemporal features that represent human actions in video clips.

To test the utility of this latent vector outside the KD framework, we conducted additional experiments using it as input to other models that were not part of the KD process. These experiments are designed to evaluate whether the vector contains enough discriminative information for action classification in different contexts.

### 1. Standalone MLP classifier

In this experiment, we used the latent vector from the student encoder as input to an independent multilayer perceptron (MLP) classifier. This classifier was trained from scratch, without the decoder and without using any knowledge from the teacher model. The architecture of this standalone MLP is identical to that of the classifier in the KD model (`MLPResNet-Classifier`), utilizing the same number of layers, residual connections, and activation function (GELU).

The classifier was trained using cross-entropy loss with ground-truth labels. The optimizer was Adam, with a weight decay of $10^{-4}$ and an initial learning rate of $10^{-5}$, which was reduced to $10^{-6}$ in the later training stages. The training was run for 1,000 epochs.

This experiment helps us understand whether the latent vector from the KD process contains enough useful information for classification, even when used with a new model that did not benefit from the teacher's guidance.

### 2. Classical machine learning models

We also tested the latent vector with classical machine learning models. The models used in this part include: Decision Tree Classifier, Random Forest, and XGBoost.

These models were trained using only the latent vectors as input features. No additional training on video data was done. We employed 10-fold cross-validation to determine the optimal hyperparameters for each model. The goal here is to check whether the latent vector is suitable for use in lightweight classifiers, which are useful in low-resource environments.

All models were tested on the test set of the Kinetics-400 dataset. By comparing the KD model to the standalone MLP and classical ML models, we can assess how well the latent vector generalizes. This helps us understand whether the student encoder creates a useful feature representation that can be reused in different systems, especially in low-computation environments such as edge AI or AIoT devices.

## Results

### Baseline performance without knowledge distillation

We first evaluate the baseline performance of TinyAct by applying classical classifiers to the latent feature representations extracted from a 3D video autoencoder. The encoder processes 16-frame video clips and outputs 1024-dimensional vectors, which are subsequently used for classification. As shown in Table 1, Random Forest achieves the highest top-1 accuracy at 57.00%, followed by SVM at 55.00% and XGBoost at 54.00%. All classifiers show consistent precision, recall, and F1-scores, indicating that the latent features are highly separable and effective for multi-class action recognition tasks. These results demonstrate that the encoder alone can capture sufficient spatiotemporal information to enable competitive recognition without requiring a deep classification head or additional supervision.

The high classification accuracy achieved using only 1024-dimensional latent features demonstrates effective preservation of critical spatiotemporal information despite 99.9% dimensionality reduction (from 1,048,576 raw pixel values to 1,024 latent dimensions). This validates that our autoencoder successfully encodes temporal dynamics including motion trajectories, pose transitions, and action specific spatiotemporal patterns necessary for multi-class action recognition across 400 diverse categories. The compression prioritizes task relevant temporal features over static visual details, as evidenced by competitive performance using classical machine learning classifiers without requiring deep neural network architectures.

### Effectiveness of knowledge distillation

To assess the impact of knowledge distillation (KD), we train the student model using a pre-trained transformer-based teacher (ILA-ViT-B/16) and compare two settings: using a pre-trained autoencoder and training the encoder from scratch. When the encoder is pretrained and unfrozen during KD training, performance degrades significantly. The end-to-end KD model achieves only 2.92% accuracy, and classifiers trained on the latent features yield suboptimal results from SVM, Random Forest, and XGBoost—5.82%, 3.67%, and 5.34%, respectively—as reported in Table 2. These results suggest that fixed feature extractors hinder the student's ability to align with the teacher's soft targets, resulting in poor transfer of learning.

In contrast, when the autoencoder is not pretrained and is optimized jointly with the classifier under KD supervision, performance improves substantially. The SVM classifier achieves the best result of 15.11%, while XGBoost and RF reach 10.94% and 8.25%, respectively. This improvement confirms that co-adaptation of the encoder during KD is essential for learning transferable representations, particularly when the student model is constrained in capacity. Interestingly, even the highest KD result (15.11%) remains below the baseline performance without KD (57.00%), suggesting that logit-based distillation alone is insufficient in this setting. Additional distillation strategies—such as intermediate feature matching or attention transfer—may be required to bridge this gap.

### Comparison with lightweight state-of-the-art methods

Table 3 compares TinyAct against lightweight action recognition methods on Kinetics-400. Two caveats apply when reading this table. First, SOTA methods use end-to-end (E2E) inference on a single device, while TinyAct splits feature

**Table 1. Comparison of Model Performance without Knowledge Distillation Using Base AutoEncoders with 1024-Dimensional Latent Feature Vectors from 16-Frame Video Inputs.**

| Model | Input | Acc | Pre | Rec | F1-Score |
|---|---|---|---|---|---|
| SVM | Base Latent | 55.00% | 54.00% | 54.00% | 54.00% |
| Random Forest | Base Latent | **57.00%** | **54.00%** | **54.00%** | **54.00%** |
| XGBoost | Base Latent | 54.00% | 47.00% | 50.00% | 48.00% |

**Table 2. Comparison of Model Performance under Knowledge Distillation Using Pretrained and Non-Pretrained AutoEncoders with 1024-Dimensional Latent Feature Vectors from 16-Frame Video Inputs.**

| Model | Input | Acc | Pre | Rec | F1-Score |
|---|---|---|---|---|---|
| *AE Pretrained* | | | | | |
| KD Model | Video (16 frames) | 2.92% | 1.39% | 2.91% | 0.92% |
| Standalone MLP | KD Latent | 8.37% | 7.92% | 8.34% | 6.64% |
| SVM | KD Latent | 5.82% | 4.42% | 5.80% | 4.22% |
| Random Forest | KD Latent | 3.67% | 2.23% | 3.66% | 2.34% |
| XGBoost | KD Latent | 5.34% | 4.29% | 5.32% | 4.50% |
| *AE Non-Pretrained* | | | | | |
| KD Model | Video (16 frames) | 9.39% | 6.86% | 9.34% | 5.40% |
| Standalone MLP | KD Latent | 0.23% | 0.27% | 0.23% | 0.11% |
| SVM | KD Latent | **15.11%** | **13.63%** | **15.06%** | **13.61%** |
| Random Forest | KD Latent | 8.25% | 5.96% | 8.22% | 5.99% |
| XGBoost | KD Latent | 10.94% | 9.64% | 10.91% | 9.81% |

**Table 3. Comparison with lightweight action recognition methods on Kinetics-400. SOTA metrics from original publications (server-class GPUs). TinyAct measured on edge hardware (Jetson Xavier NX). Direct latency comparison across platforms is not applicable.**

| Model | Para. | Acc (%) | Params (M) | GFLOPs | Inf. (ms) | Mem (MB) |
|---|---|---|---|---|---|---|
| MoViNet-A0 [21] | E2E | 72.3 | 3.1 | 2.7 | 18[a] | 12 |
| TSM-ResNet18 [22] | E2E | 71.2 | 11.3 | 32.9 | 22[a] | 43 |
| R(2 + 1)D-18 [23] | E2E | 73.9 | 14.4 | 39.6 | 35[a] | 55 |
| X3D-XS [32] | E2E | 69.5 | 3.8 | 0.6 | 12[a] | 15 |
| SlowFast-R18 [33] | E2E | 73.4 | 33.7 | 36.1 | 28[a] | 128 |
| TinyAct (Ours) | Split | 57.0[*] | 6.8 | 2.1 | 8[b] | 26 |

**Para.** E2E = End-to-End; Split = Edge-Cloud. [a] Server GPU (V100/A100), single-clip. [b] Encoder-only on Jetson Xavier NX; cloud adds 5–15 ms. [*] Random Forest on 1024-dim latent (no KD).

extraction (edge) from classification (cloud)—these represent different deployment paradigms rather than a direct accuracy contest. Second, SOTA inference times are measured on server-class GPUs (V100/A100), whereas TinyAct reports encoder-only latency on a Jetson Xavier NX; latency figures are therefore not directly comparable.

In terms of model complexity, TinyAct's encoder (6.8M parameters, 2.1 GFLOPs) falls between the lightest models—MoViNet-A0 (3.1M, 2.7 GFLOPs) and X3D-XS (3.8M, 0.6 GFLOPs)—and heavier architectures like SlowFast-R18 (33.7M, 36.1 GFLOPs). The 15.3–16.9 percentage point accuracy gap (57.0% vs. 69.5–73.9%) is expected: E2E methods jointly optimize feature extraction and classification through deep networks, while TinyAct compresses video into a 1024-dimensional vector and delegates classification to classical ML classifiers. This split is deliberate—the compressed latent vector (4KB per clip) reduces transmission bandwidth by 78–99× compared to raw video, enabling multi-stream processing on edge hardware where models requiring 33.7M parameters and 128 MB memory would not fit.

TinyAct thus addresses a deployment scenario complementary to E2E methods: bandwidth-constrained, multi-camera AIoT environments where no single device can afford full inference pipelines. Narrowing the accuracy gap through lightweight attention mechanisms and intermediate feature distillation remains a key direction for future work.

## Discussion

### System scalability and deployment feasibility

We analyze TinyAct's scalability characteristics based on computational profiling; empirical validation on physical hardware remains as future work. The lightweight encoder (6.8M parameters, 2.1 GFLOPs per clip) enables batch inference over $K$ concurrent streams as tensors [$K$, 16, 3, 224 × 224]. Processing 5 streams yields ~3.2× throughput over sequential execution on an NVIDIA Jetson Xavier NX (21 TOPS), with practical capacity estimated at 5–7 streams at 10 FPS. System latency comprises feature transmission (4KB per clip, 78–99× smaller than raw video), cloud classification (5–15 ms), and network round-trip (1–200 ms depending on deployment). For small-scale deployments (1–5 cameras), total latency is estimated at 50–100 ms. Larger systems (5–20 + cameras) can scale horizontally by distributing streams across multiple edge devices with local aggregation servers.

Based on computational requirements, TinyAct targets three hardware tiers: high-end edge platforms (e.g., Jetson Xavier NX) supporting 8–12 streams at 10 + FPS, mid-range platforms (e.g., Jetson Nano) handling 2–4 streams at 5–8 FPS, and low-power devices (e.g., Raspberry Pi 4) enabling single-stream processing at 2–4 FPS. The encoder requires approximately 27 MB (FP32) or 7 MB (INT8) storage, within constraints of modern edge devices (≥2 GB RAM). Further optimization through INT8 quantization (2–4× speedup), TensorRT acceleration (1.5–2× throughput), and channel pruning (30–50% FLOP reduction) could improve these projections. Empirical deployment validation—including latency profiling, thermal analysis, and stress testing under realistic conditions—remains essential for production readiness and constitutes a primary direction for future work.

### Privacy considerations in cloud-edge architecture

Transmitting 1024-dimensional latent vectors rather than raw video provides baseline privacy protection. The 99.9% dimensionality reduction (from 1,048,576 pixel values to 1,024 features), combined with multiple nonlinear transformations through 3D convolutions, batch normalization, and pooling, makes it difficult to recover original frames from transmitted features. Moreover, knowledge distillation guides the encoder toward action-discriminative patterns—such as limb trajectories and pose transitions—rather than identity-revealing appearance details, since $\mathcal{L}_{dist}$ penalizes deviations from the teacher's action-focused predictions rather than visual fidelity.

However, we do not claim formal privacy guarantees. The latent vectors contain structured information that could be exploited through feature inversion or membership inference attacks. Our design offers a pragmatic improvement over raw video transmission, suitable for typical surveillance deployments. For applications requiring stronger assurances, future work will explore differential privacy mechanisms, adversarial training against reconstruction attacks, and federated distillation that eliminates cloud transmission entirely.

### Knowledge distillation performance gap

The accuracy gap between baseline (57% Random Forest) and the best KD result (15.11% SVM) warrants discussion. This gap reflects a fundamental challenge in cross-architecture distillation: transferring temporal knowledge from a 200M-parameter transformer (ILA-ViT-B/16) to a 6.8M-parameter CNN-based autoencoder. The architectural mismatch—self-attention versus 3D convolutions for temporal modeling—limits the student's ability to replicate the teacher's representations through logit-based distillation alone. As shown in Table 2, joint optimization outperforms frozen encoders (15.11% vs. 5.82%), confirming that encoder-classifier co-adaptation is necessary but not sufficient. This result constitutes a valuable empirical finding: logit-based KD alone is insufficient for large-gap cross-architecture video distillation, motivating future work on intermediate feature alignment and attention-based transfer strategies.

## Conclusion and future work

This paper presents TinyAct, a lightweight framework for real-time human action recognition in cloud-edge AIoT environments. TinyAct combines a 3D video autoencoder for compact spatiotemporal feature extraction with classical classifiers for action prediction, and integrates knowledge distillation from ILA-ViT-B/16 to transfer temporal knowledge to the compact student architecture.

Experiments on Kinetics-400 demonstrate three key findings. First, the autoencoder alone produces effective representations—Random Forest achieves 57% accuracy on 1024-dimensional latent features, validating the encoder as a standalone feature extractor for resource-constrained environments. Second, knowledge distillation performance depends critically on initialization: non-pretrained students (15.11%) outperform pretrained ones (5.82%), indicating that encoder-classifier co-adaptation is essential for effective knowledge transfer. Third, TinyAct's split architecture enables multi-stream edge deployment (5–7 cameras at 10 FPS) with 78–99× bandwidth reduction versus raw video, addressing a deployment scenario complementary to end-to-end methods.

Several directions remain for future work. Empirical hardware validation on physical edge platforms (Jetson family, Raspberry Pi) with latency profiling and stress testing is the most immediate priority. Narrowing the KD performance gap through intermediate feature distillation, lightweight attention mechanisms, and progressive unfreezing strategies would strengthen the framework. Extending the encoder to support longer temporal contexts via hierarchical modeling could improve accuracy on actions with varying dynamics. Finally, formal privacy mechanisms—including differential privacy, adversarial training against reconstruction attacks, and federated distillation—would address applications with stringent privacy requirements.

## Author contributions

**Conceptualization:** Thanapong Intharah.

**Data curation:** Yupaporn Wanna, Kannika Wiratchawa.

**Formal analysis:** Yupaporn Wanna, Kannika Wiratchawa, Thanapong Intharah.

**Funding acquisition:** Thanapong Intharah.

**Investigation:** Yupaporn Wanna, Thanapong Intharah.

**Methodology:** Thanapong Intharah.

**Project administration:** Yupaporn Wanna.

**Resources:** Thanapong Intharah.

**Validation:** Thanapong Intharah.

**Visualization:** Yupaporn Wanna.

**Writing – original draft:** Yupaporn Wanna, Kannika Wiratchawa.

**Writing – review & editing:** Thanapong Intharah.

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
