## [Decision Letter · Decision Letter 0]

27 Aug 2025

Dear Dr. Intharah,

Thank you for submitting your manuscript to PLOS ONE. After careful consideration, we feel that it has merit but does not fully meet PLOS ONE’s publication criteria as it currently stands. Therefore, we invite you to submit a revised version of the manuscript that addresses the points raised during the review process.

We look forward to receiving your revised manuscript.

Kind regards,

Muhammad Bilal

Academic Editor

PLOS ONE

Journal Requirements:

This study was financially supported by the Fundamental Fund of Khon Kaen University (fiscal year 2024, project number 200553 ), the National Science, Research and Innovation Fund (NSRF), Thailand.

This study was financially supported by the Fundamental Fund of Khon Kaen University (fiscal year 2024, project number 200553 ), the National Science, Research and Innovation Fund (NSRF), Thailand.

This study was financially supported by the Fundamental Fund of Khon Kaen University (fiscal year 2024, project number 200553 ), the National Science, Research and Innovation Fund (NSRF), Thailand.

5. We note that Figure 1 includes an image of a [patient / participant / in the study].

Reviewers' comments:

Reviewer's Responses to Questions

**Comments to the Author**

1. Is the manuscript technically sound, and do the data support the conclusions?

Reviewer #1: Yes

Reviewer #2: Yes

2. Has the statistical analysis been performed appropriately and rigorously?

Reviewer #1: N/A

Reviewer #2: Yes

3. Have the authors made all data underlying the findings in their manuscript fully available?

Reviewer #1: Yes

Reviewer #2: Yes

4. Is the manuscript presented in an intelligible fashion and written in standard English?

Reviewer #1: Yes

Reviewer #2: Yes

Reviewer #1: The idea is good but there is a major room for further improvement. Plz address the following issues before next round.

Comment 1: The paper presents a compelling solution to the critical challenge of deploying real-time action recognition on resource-constrained edge devices. The combination of edge computing with cloud-based processing through knowledge distillation represents a practical approach to balance computational efficiency with accuracy requirements. Please take assistance from the study titled “efficient deepfake detection via layer-frozen assisted dual attention network for consumer imaging devices” for optimizing lightweight architectures on consumer-grade edge devices.

Comment 2: The AIoT architecture design is well-structured, clearly separating edge device responsibilities (data collection, preprocessing, feature extraction) from cloud processing (action classification, storage, visualization). This separation enables effective resource utilization while maintaining real-time processing capabilities. Please take assistance from the study titled “lightweight transformer-driven multi-scale trapezoidal attention network for saliency detection” for implementing efficient attention mechanisms in the edge processing pipeline.

Comment 3: The knowledge distillation approach using ILA (Intermediate Layer Alignment) as a teacher model to transfer temporal knowledge to the compact student architecture is technically sound. However, the paper lacks detailed explanation of how the distillation process handles the temporal complexity inherent in action recognition tasks. Please take assistance from the study titled “optimal features driven hybrid attention network for effective video summarization” for better temporal feature extraction and knowledge transfer strategies.

Comment 4: The experimental validation on Kinetics-400 dataset demonstrates competitive performance (57.00% accuracy) compared to larger models, but the evaluation appears limited to a single dataset. Testing on additional action recognition benchmarks (UCF-101, HMDB-51) is necessary to establish generalizability. Please take assistance from the study titled “bilateral feature fusion with hexagonal attention for robust saliency detection under uncertain environments” for handling diverse environmental conditions and dataset variations.

Comment 5: The 16-frame video clip processing with 1024-dimensional latent features shows promise for computational efficiency, but the paper does not provide sufficient analysis of how frame sampling strategies affect recognition accuracy for different action types with varying temporal dynamics. Please take assistance from the study titled “optimal features driven hybrid attention network for effective video summarization for developing adaptive frame sampling and temporal attention mechanisms.

Comment 6: The comparison with SVM (55.00%) and XGBoost (54.00%) baselines provides context, but the paper lacks comparison with other state-of-the-art lightweight action recognition methods specifically designed for edge deployment, limiting the assessment of relative performance. Please take assistance from the study titled “efficient deepfake detection via layer-frozen assisted dual attention network for consumer imaging devices” for comparative analysis with other efficient detection frameworks on edge devices.

Comment 7: The modular architecture design enabling flexible deployment across diverse hardware configurations is a significant strength. However, the paper does not provide concrete examples of deployment on specific edge devices or performance metrics on actual hardware platforms. Please take assistance from the study titled “lightweight transformer-driven multi-scale trapezoidal attention network for saliency detection” for practical deployment strategies on resource-constrained devices.

Comment 8: The integration of privacy-preserving data management through local edge processing is valuable for surveillance applications, but the paper does not address potential privacy concerns when compressed features are transmitted to the cloud for classification. Please take assistance from the study titled “bilateral feature fusion with hexagonal attention for robust saliency detection under uncertain environments” for developing robust feature representations that maintain privacy while preserving essential information.

Comment 9: The real-time monitoring interface and dashboard visualization components enhance practical applicability, but the paper lacks discussion of system scalability when handling multiple concurrent video streams or network latency effects on performance. Please take assistance from the study titled “efficient deepfake detection via layer-frozen assisted dual attention network for consumer imaging devices” for handling multiple stream processing efficiently on consumer devices.

Comment 10: The overall contribution advances the field by demonstrating that effective human action recognition can be achieved without computationally intensive deep networks on edge devices. However, the evaluation needs strengthening through more comprehensive hardware testing and broader dataset validation. Please take assistance from the study titled “lightweight transformer-driven multi-scale trapezoidal attention network for saliency detection” for comprehensive evaluation methodologies on lightweight architectures.

Technical Questions:

Question 1: How does the knowledge distillation process specifically handle temporal dependencies during the teacher-to-student knowledge transfer, and what mechanisms ensure that critical spatiotemporal features are preserved in the compressed representation? Please take assistance from the study titled “optimal features driven hybrid attention network for effective video summarization” for temporal dependency modeling.

Question 2: What is the impact of network latency and bandwidth limitations on the system performance when transmitting compressed features from edge devices to the cloud, and how does the system handle situations where cloud connectivity is intermittent or unavailable? Please take assistance from the study titled “bilateral feature fusion with hexagonal attention for robust saliency detection under uncertain environments” for developing robust communication strategies under uncertain network conditions.

Reviewer #2: Citation [1] in the first line of introduction makes no sense.

No need to write how authors validate their model in the introduction section it should be part of results section.

Author must write about menuscript breakdown in last paragraph of introduction like section 2 reperesents literature review section 3 represents methodology etc.

Author must include more details in Transformer-Based Architectures for Action Recognition, Self-Supervised Learning for Action Recognition otherwise dont write them as heading and convert all data in simple paragraph style.

Unbold Knowledge Distillation (KD) on line 133.

Author should merge Training Procedure and Hyperparameter Settings in Training Configuration and Implementation Details because both looks same otherwise specify the difference between then and need to write them in two different places.

Authors mentioned results of their own experimentations and did not perform SOTA comparison in this study which needs to include.

.

Reviewer #1: No

Reviewer #2: No

---

## [Author Response · Author response to Decision Letter 1]

3 Mar 2026

The responses are in the response to reviewers file.

---

## [Decision Letter · Decision Letter 1]

30 Mar 2026

TinyAct: A framework for Real-time Action Recognition in the Cloud through Distillation Learning

PONE-D-25-38260R1

Dear Dr. Intharah,

We’re pleased to inform you that your manuscript has been judged scientifically suitable for publication and will be formally accepted for publication once it meets all outstanding technical requirements.

Kind regards,

Neng Ye

Academic Editor

PLOS One

Additional Editor Comments (optional):

The authors have addressed all the comments. I'm OK with this version.

Reviewers' comments:

Reviewer's Responses to Questions

**Comments to the Author**

Reviewer #1: All comments have been addressed

Reviewer #2: All comments have been addressed

2. Is the manuscript technically sound, and do the data support the conclusions?

Reviewer #1: Yes

Reviewer #2: Yes

3. Has the statistical analysis been performed appropriately and rigorously?

Reviewer #1: N/A

Reviewer #2: Yes

4. Have the authors made all data underlying the findings in their manuscript fully available?

Reviewer #1: Yes

Reviewer #2: Yes

5. Is the manuscript presented in an intelligible fashion and written in standard English?

Reviewer #1: Yes

Reviewer #2: Yes

Reviewer #1: Well done authors for addressing all the comments and constructive revision. I recommend this article for acceptance

Reviewer #2: (No Response)

.

Reviewer #1: No

Reviewer #2: No

---

## [Editor Report · Acceptance letter]

PONE-D-25-38260R1

PLOS One

Dear Dr. Intharah,

I'm pleased to inform you that your manuscript has been deemed suitable for publication in PLOS One. Congratulations! Your manuscript is now being handed over to our production team.

Kind regards,

on behalf of

Dr. Neng Ye

Academic Editor

PLOS One